Corrected: Publisher correction

# Prominence of the tropics in the recent rise of global nitrogen pollution

Minjin Lee[1], Elena Shevliakova[2], Charles A. Stock[2], Sergey Malyshev[2] & P.C.D. Milly[3]

Nitrogen (N) pollution is shaped by multiple processes, the combined effects of which remain uncertain, particularly in the tropics. We use a global land biosphere model to analyze historical terrestrial-freshwater N budgets, considering the effects of anthropogenic N inputs, atmospheric $CO_2$, land use, and climate. We estimate that globally, land currently sequesters 11 (10–13)% of annual N inputs. Some river basins, however, sequester >50% of their N inputs, buffering coastal waters against eutrophication and society against greenhouse gas-induced warming. Other basins, releasing >25% more than they receive, are mostly located in the tropics, where recent deforestation, agricultural intensification, and/or exports of land N storage can create large N pollution sources. The tropics produce 56 ± 6% of global land N pollution despite covering only 34% of global land area and receiving far lower amounts of fertilizers than the extratropics. Tropical land use should thus be thoroughly considered in managing global N pollution.

[1] Program in Atmospheric and Oceanic Sciences, Princeton University, 300 Forrestal Road, Sayre Hall, Princeton, NJ 08540, USA. [2] NOAA/Geophysical Fluid Dynamics Laboratory, 201 Forrestal Road, Princeton, NJ 08540, USA. [3] U.S. Geological Survey and NOAA/Geophysical Fluid Dynamics Laboratory, 201 Forrestal Road, Princeton, NJ 08540, USA. Correspondence and requests for materials should be addressed to M.L. (email: minjinl@princeton.edu)

Anthropogenic reactive nitrogen (Nr) inputs to terrestrial ecosystems (e.g., synthetic fertilizers and atmospheric deposition associated with agricultural industrialization and fossil fuel combustion) and subsequent (N) losses to the ocean and atmosphere have sharply increased over the past century[1,2]. N fluxes to the coastal ocean fuel eutrophication[3], increase harmful algal blooms[4], and expand hypoxic dead zones[5]. Nitrous oxide emitted from soils, rivers, and lentic systems are the third largest greenhouse gas contributor to radiative forcing[6] and the dominant ozone-depleting factor[7].

Much attention has been placed on the effects of sharply increased anthropogenic Nr inputs on severe oceanic or atmospheric pollution[5,8–10]. However, numerous other anthropogenic and natural processes modulate N fluxes to the ocean and atmosphere. Biological N fixation (BNF) adds substantial Nr to terrestrial ecosystems, especially in the tropics[1,11,12] and agricultural systems with intensive legume cultivation[13]. Land use and land cover change (LULCC) export substantial N and carbon (C) from storage in wood, crops, grasses, and soils via deforestation, slash-and-burn agriculture, harvesting, and livestock grazing[6,14–17]. In some regions, increasing terrestrial Nr inputs directly augment N fluxes to the ocean and atmosphere[10,18,19], whereas terrestrial ecosystems in other regions sequester a significant fraction of added Nr, buffering N fluxes[19,20]. This regional variation in the capacity of terrestrial ecosystems to sequester Nr inputs appears to depend on both current and past anthropogenic land disturbances[20] and changing climate and variability[21].

The fate of Nr in many terrestrial and freshwater ecosystems remains uncertain because of sparse measurements of N cycling processes[1,22,23]. Such uncertainty remains a major research and management challenge, particularly for tropical regions, which are expected to experience the most dramatic increases in anthropogenic Nr inputs and LULCC in the next few decades[24,25]. Furthermore, increasing anthropogenic Nr inputs to relatively N-rich tropical systems may result in more rapid N losses to the ocean and atmosphere than to N-limited temperate systems[26].

Here we used the global land biosphere model LM3-TAN[27] which represents coupled water, C, and N dynamics and interactions within a vegetation–soil–river–lake system and simulates the exchanges between and transformations within each subsystem for three N species (i.e., organic, ammonium, and nitrate plus nitrite N)[27–30]. See Methods for a model description. Following ~11,000 years of spin-up, LM3-TAN simulated global land N storage and fluxes to the ocean and atmosphere from 1700 to 2005 when forced by historical changes in atmospheric $CO_2$, anthropogenic Nr inputs[8,9,31], climate[32], and LULCC[15] at $1 \times 1$ degree resolution. See Methods for model forcing and simulations.

We analyze the past two and half centuries of terrestrial and freshwater N storage and fluxes to the ocean and atmosphere, considering not only the effect of increased anthropogenic Nr inputs, but also the effects of elevated atmospheric $CO_2$ (i.e., $CO_2$ fertilization), LULCC, and climate change. We demonstrate how LULCC has disrupted vegetation-soil systems across 159 major river basins covering 70% of global land area. We then show how this has altered terrestrial-freshwater N cycling and affected N fluxes and pollution to the ocean and atmosphere at global scales, across the tropics (between 23.26°N and 23.26°S) and extratropics, and at the basin scale. These analyses suggest that recent accelerating LULCC in tropical regions has resulted in prominence of the tropics in global N pollution despite their disproportionately smaller land area and far lower fertilizer applications than the extratropics.

## Results

**Evaluation of global and regional N and C budgets.** LM3-TAN simulated land N storage in vegetation, soils, litter, rivers, and lakes (Fig. 1). Total land Nr inputs are the sum of simulated BNF[28], synthetic fertilizers[8], and atmospheric deposition[9]. Total land N outputs are the sum of river exports to the ocean[27], emissions to the atmosphere[27,28], and net harvest – N in harvested woods, crops, and grasses[30] after subtracting out manure applied to croplands[8] and urban wastewater discharges[31]. The majority of the net harvest is assumed to ultimately go to the atmosphere via various pathways including wood, biofuel, and waste burning, livestock respiration, emissions from food, human, and livestock waste[1,6,8], though some is sequestered in durable goods (i.e., home building). See Methods for a detailed description of the input and output terms and Supplementary Table 1 and Supplementary Note 1 for further discussion.

Simulated global land N storage and fluxes in LM3-TAN are found to be within published uncertainty bounds in 16 different studies, when comparable categorization, definitions, and assumptions are applied (Fig. 1, Supplementary Table 1, Supplementary Note 1). Simulated regional river discharge, dissolved inorganic and organic N loads and concentrations also agree with reported discharge and measurement-based N estimates from 47 major rivers, which are distributed broadly over the globe and influenced by various climates, biomes, and human activities (Supplementary Figures 1 and 2, Supplementary Table 2). The global C balance is also found to be generally within uncertainty bounds in large-scale constraints and atmospheric studies. Simulated global net land C fluxes (e.g., 1.0 (0.9–1.3) PgC $yr^{-1}$ for 1990s, 1.1 (1.0–1.3) PgC $yr^{-1}$ for 2001–2004) agree with estimates based on global C budget constraints (e.g., 1.1 (0.5–1.8) PgC $yr^{-1}$ for 1990s)[33] and inverse models (e.g., 0.3–1.7 PgC $yr^{-1}$ for 2001–2004)[34,35]. The simulated land-use change contribution to elevated $CO_2$ between 1750 and 2005 is 217 PgC, within a published range of $180 \pm 80$ PgC between 1750 and 2011[6]. The simulated cumulative net land C source between 1750 and 2005 is 99 (65–107) PgC, consistent in magnitude, albeit at the upper end of reported uncertainty ranges of $30 \pm 45$ PgC between 1750 and 2011[6].

**Global transition of land from a net N source to sink.** Globally, our results suggest that, from 1750 until the late 1940s, land served as a net N source (Fig. 2a): reduced land N storage (cyan line) augmented N outputs from land (orange line) such that they exceeded contemporaneous Nr inputs to land (black line). That is, substantial exports of legacy land N storage enhanced N fluxes to the ocean and atmosphere relative to what would had been caused by the contemporaneous Nr inputs alone. Since the late 1940s, land has become a net N sink: land systems have acted to reduce N outputs to the ocean and atmosphere by sequestrating a fraction of contemporaneous Nr inputs. The global transition of land from a net N source to a net N sink in the late 1940s mirrors a similar transition in terrestrial C storage[33]. It is a robust result of increasing N demand from vegetation and subsequent N accumulation in soils associated with secondary forest regrowth in some regions and overall enhanced vegetation growth due to $CO_2$ fertilization[35–39]. Similar transitions also occur in all of our N cycle sensitivity simulations that were forced by different BNF settings[1,9], fertilizer inputs[40], and LULCC[15] (Supplementary Figure 3). See Methods for a description of the baseline and sensitivity simulations. Removing the $CO_2$ fertilization effect delays the global land transition from a source to sink, but a marked reduction in the net N source is still apparent by the late 20th century even without $CO_2$ fertilization (Fig. 2b).

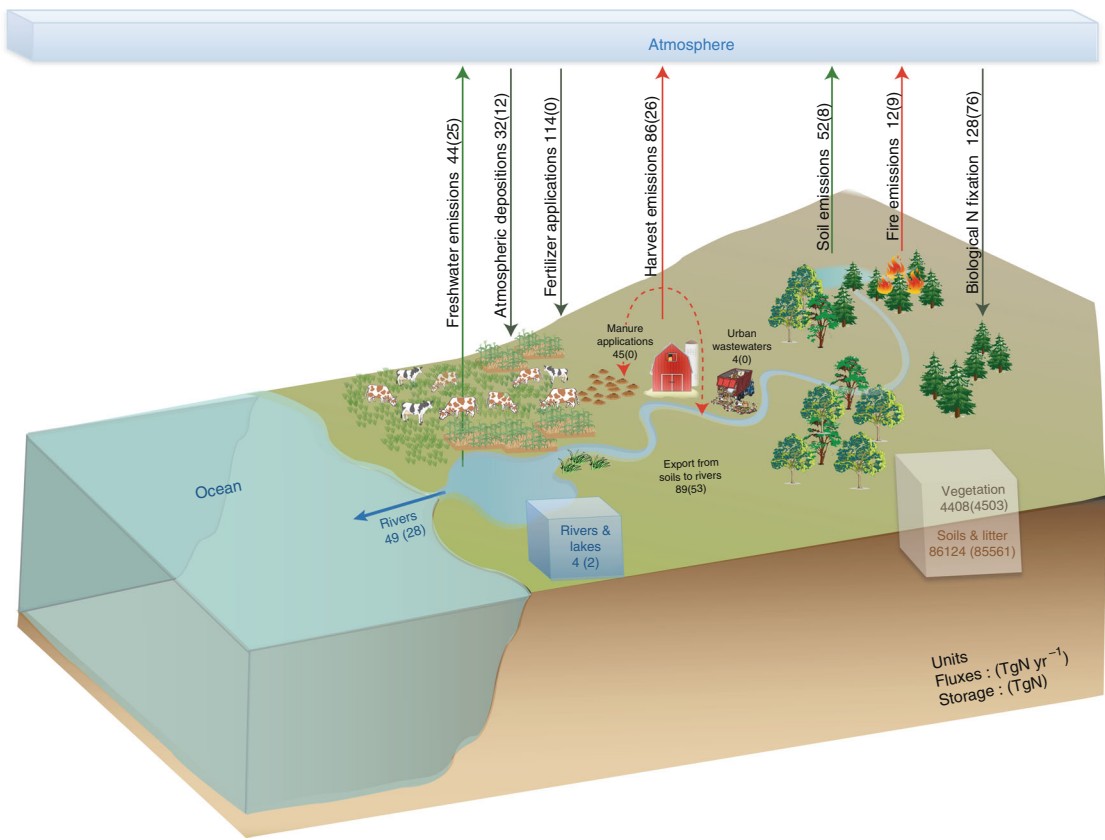

**Fig. 1** Global N budget. Numbers represent global land N storage in TgN or annual N exchange fluxes in TgN yr$^{-1}$ for contemporary (1991–2005 average) and preindustrial (1831–1860 average in parenthesis) times. These results are summarized, discussed, and compared with reported estimates from various scientific studies in Supplementary Table 1 and Supplementary Note 1

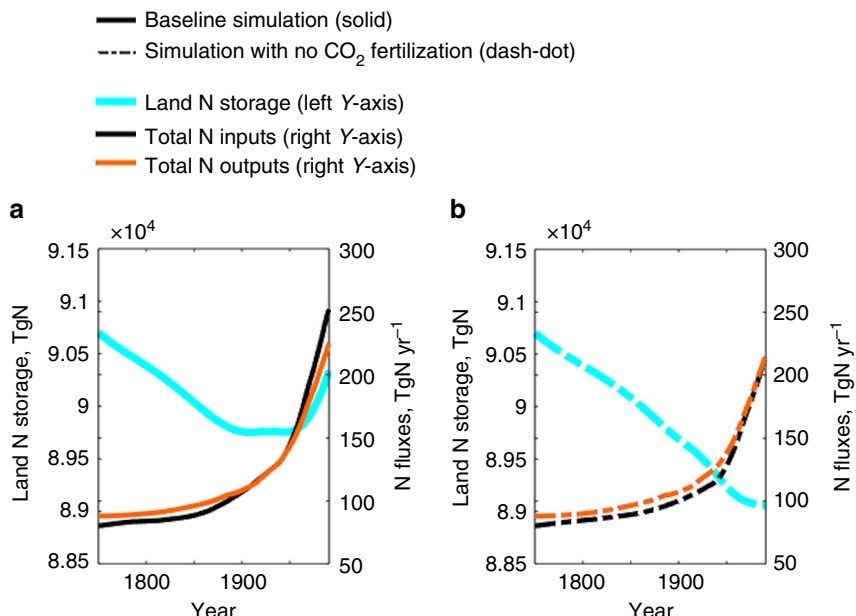

**Fig. 2** Global land N storage and fluxes under different scenarios. **a** The baseline simulation (solid lines). **b** The no $CO_2$ fertilization simulation (dash-dot lines). The colors show land N storage (cyan), total land Nr inputs (black), and total land N outputs (orange). All plots show 30-year moving averages from 1750 to 2005

To estimate the evolving land capacity to sequester Nr inputs, we define a global and basin-specific N-Loss Index, NLI, as total N outputs from land, divided by total Nr inputs to land. NLI larger than 1 indicates that land is releasing an amount of N that exceeds total land Nr inputs, decreasing land N storage and amplifying N outputs. NLI smaller than 1 indicates that land is sequestrating a fraction of total land Nr inputs, increasing land N storage and releasing less N than it receives. When NLI is 1, land inputs and outputs are in balance. In a manner consistent with Fig. 2a, global NLI is between 1–1.1 up to the late 1940s, reflecting 0–10% augmentation of total land Nr inputs (Fig. 3a). It then falls below 1 and decreases, eventually sequestrating 11% of total land Nr inputs. This result is robust across the different BNF settings, fertilizer inputs, and LULCC (Fig. 3a).

**Zonal variation in land N sequestration or release**. When NLI is broken down by latitude, results suggest that tropical NLI is generally higher than extratropical NLI over the last century (Fig. 3a, magenta vs. cyan lines). By the latter half of the 20th century, the extratropics become a strong net N sink, sequestering 18 (18–20)% of total land Nr inputs, while the tropics become nearly neutral (NLI = ~1) despite the global tendency toward net N sequestration. These tropical systems ultimately produce 48 (46–69)% of global land N outputs to the ocean and atmosphere (Table 1).

The total input and output fluxes comprising the NLI allow us to assess whether land is sequestrating or releasing N at global and basin scales. Estimating N pollution from land, however, requires additional considerations. Partitioning of the N outputs into pollutants and environmentally benign forms (see Methods

and Supplementary Note 2) further suggests that the tropics produce 56 ± 6% of global land N pollution to the ocean and atmosphere (Fig. 4). Even without $CO_2$ fertilization, tropical contributions to global land N pollution are 57 ± 6% (Supplementary Figure 4). These high contributions occur despite the tropics covering only 34% of global land area and receiving much lower amounts of synthetic fertilizers than the extratropics (Table 1, Fig. 5a, b).

In the tropics, the largest contributors to increasing land N outputs are net harvest and denitrification (Fig. 5c). As mentioned previously, the majority of the net harvest is assumed to ultimately go to the atmosphere[1,6,8]. Most of the increasing tropical N outputs thus go to the atmosphere, and river exports to the ocean remain relatively stable. Averaged across all tropical systems, the increasing N outputs are in near balance with increasing agricultural BNF (largely due to expansion of agricultural land areas) and, more recently, limited increases in fertilizer inputs and atmospheric deposition (Fig. 5a), explaining the nearly neutral N fluxes to land storage (Fig. 5e). In the extratropics, contemporary (1976–2005 mean) total land Nr inputs amount to 136 TgN yr$^{-1}$ (Fig. 5b), however, enhanced land N sequestration (Fig. 5f) reduces total land N outputs to 111 TgN yr$^{-1}$ (Fig. 5d). Rapid increases in fertilizer use and relatively modest increases in atmospheric deposition and agricultural BNF have continued to augment N outputs despite the enhanced N sequestration. It is notable that in the extratropics, the increasing N outputs not only go to the atmosphere, but also are exported to the ocean in the form of bioavailable inorganic N. The same essential source/sink dynamics evident in Fig. 5 operate for the other BNF settings, fertilizer inputs, and LULCC, with only modest changes in relative importance (Supplementary Figures 5–8).

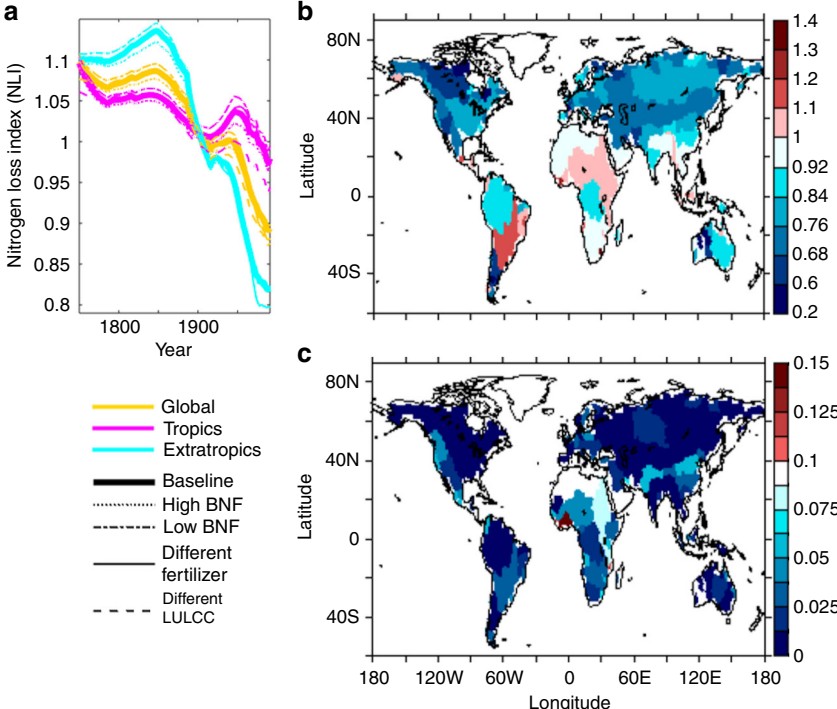

**Fig. 3** N-Loss Index (NLI). **a** Global (yellow), tropical (magenta), and extratropical (cyan) NLIs. The thick line shows the baseline simulation. The thin lines show results forced by different inputs: dot (high BNF), dash-dot (low BNF), solid (Lu and Tian's fertilizer[40]), and dash (LULCC scenario without shifting cultivation[15]). Globally, the baseline simulation produces 128 TgN yr$^{-1}$ of BNF (medium) during 1991–2005 that is between 116 TgN yr$^{-1}$ (low) and 145 TgN yr$^{-1}$ (high) (Supplementary Table 1 and Supplementary Note 1). The low BNF is close to an estimate by Green and colleagues[9] (112 TgN yr$^{-1}$) and the high BNF is close to an estimate by Galloway and colleagues[1] (139 TgN yr$^{-1}$). The NLIs were calculated after the results were 30-year moving averaged from 1750 to 2005. **b, c** Contemporary (1976–2005 average) NLIs for 159 major river basins (**b**) and their standard deviation across the different BNF settings, fertilizer inputs, and LULCC (**c**)

**Table 1 Tropical and global land area and N outputs**

|  | Land area | River DON exports | River DIN exports | Fire emissions | Denitrification emissions | Net harvest exports | Total land N outputs | Total land N outputs, % |
|---|---|---|---|---|---|---|---|---|
| Tropics | 51 | 10 | 12 (11–13) | 7 | 36 (27–37) | 52 (42–61) | 117 (107–127) | 48 (46–49) |
| Global | 151 | 17 (17–18) | 32 (24–32) | 12 | 96 (73–98) | 86 (76–98) | 242 (217–257) | 100 |

Numbers in parentheses show uncertainty test results, with a range of BNF settings[1,9], different fertilizers[40], different LULCC (without shifting cultivation)[15], and different fractionation of N species in Nr inputs[63,64]. See Methods for model forcing and simulations. Units are$10^6$ km$^2$ for land area and TgN yr$^{-1}$ for fluxes

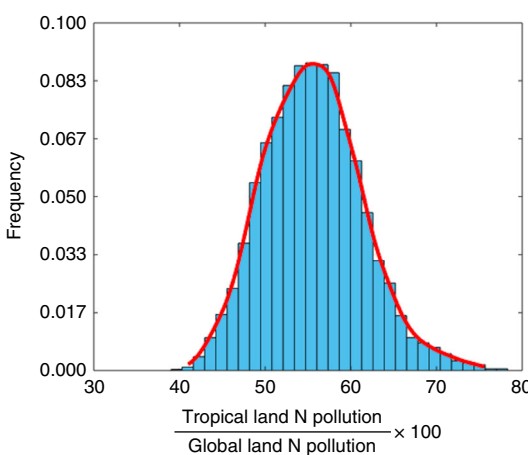

**Fig. 4** Prominence of the tropics in global N pollution. Tropical (global) land N pollution was estimated as the sum of N outputs from tropical (global) land, minus the sum of environmentally benign portions of the outputs (see Methods and Supplementary Note 2). These estimates suggest that the tropics create 56 ± 6% of global land N pollution despite covering only 34% of global land area and receiving far lower amounts of synthetic fertilizers than the extratropics (Table 1, Fig. 5a, b). The reported uncertainties consider the sensitivity of the pollution estimates to variations in the forcing (i.e., different BNF settings, fertilizer inputs, and LULCC) and different partitioning of the outputs into environmentally benign vs. pollutant forms

In the no-$CO_2$ fertilization scenario, since the late 19th century, agricultural and non-agricultural BNF in the tropics and extratropics are lower than those in the baseline simulation (Fig. 5a, b, dash-dot vs. solid lines). The lower BNF decreases land N storage, primarily in tropical non-agricultural lands including large intact forests, but does not immediately reduce overall N outputs (Fig. 5c–f). Since the mid 20th century, the lower BNF results in only a small reduction in N outputs, specifically agricultural net harvest. Thus, the comparison of simulations with and without $CO_2$ fertilization suggests that $CO_2$ fertilization has increased land N storage primarily in undisturbed areas, but have not significantly affected tropical and extratropical N outputs.

**Basin-scale variation in land N sequestration or release.** Inspection of basin-scale NLI patterns (Fig. 3b) shows that considerable variation in land N sequestration or release underlie the emergent global and latitudinal patterns discussed above. Contemporary basin-specific NLIs vary between substantial sinks (<0.5) and substantial sources (>1.25). That is, basins with NLI much smaller than 1 are providing a valuable ecosystem service by sequestering a significant fraction of their total Nr inputs, whereas other basins are releasing an amount of N that far exceeds the sum of their Nr inputs, amplifying N fluxes to the atmosphere or ocean. This large regional variation is consistent with N isotope records[10,19,20] and aligns with bottom-up estimates of C fluxes in forests, with different regions, countries,

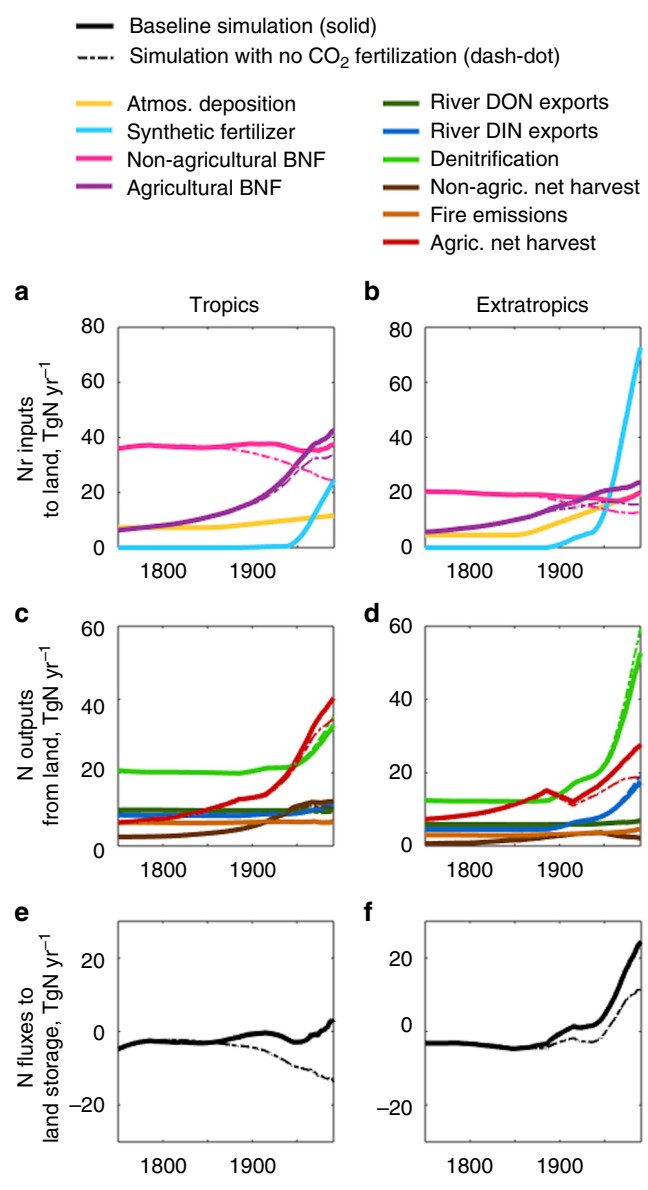

**Fig. 5** Land N fluxes in the tropics and extratropics. **a, b** Land Nr inputs include atmospheric deposition (light orange), synthetic fertilizers (sky-blue), biological N fixation (BNF) in non-agricultural (plum) and agricultural (purple) lands. **c, d** Land N outputs include river dissolved organic N (DON) exports (green), river dissolved inorganic N (DIN) exports (blue), soil and freshwater denitrification (light green), fire emissions (orange), net harvest in agricultural (red) and non-agricultural (brown) lands. **e, f** N fluxes to land storage. Solid thick lines show the baseline simulation. Dash-dot thin lines show the no-$CO_2$ fertilization simulation. All plots show 30-year moving averages from 1750 to 2005

biomes, or land disturbances displaying distinctive patterns in C gain or loss[37,41]. Basins with NLIs smaller than 1 are far more prevalent in temperate regions, while those with NLIs exceeding 1 or nearly 1 are more prevalent in the tropics, but exceptions exist. The standard deviation in NLIs between the different BNF settings, fertilizer inputs, and LULCC is generally <0.05 (Fig. 3c), and 4 or more of the 5 simulations agree on whether a basin is a net N sink or source during 1976–2005 for all but 5 of the 159 basins considered.

Analysis of evolving land N storage and fluxes at the basin scale suggests that the primary driver of variation in land N sequestration or release is the timing, intensity, and legacy of prior agricultural land use and deforestation (i.e., LULCC footprints). We classify the 159 basins based on five small-to-large evolving types of LULCC footprints, which are here reflected by five distinct basin-scale pathways of land N storage from 1750 to 2005 (Fig. 6).

Small footprint basins (Fig. 6a, f; e.g., Mackenzie) have been modestly influenced by slow, minimal, and late agricultural land use and deforestation (see Supplementary Figure 9 for historical land-use changes and Supplementary Figure 10 for global distributions of contemporary land use). These basins are

furthermore located at high-latitude boreal forest and tundra regions and exhibit slow N accumulation, reflecting long equilibrium timescales of the systems[42] (See Methods). These factors combine with modest secondary forest regrowth and enhanced vegetation growth due to $CO_2$ fertilization to yield persistent increases in land N storage.

Small-to-medium footprint basins (Fig. 6b, f; e.g., Colorado) are characterized by relatively slow, mild, and late LULCC, but changes are considerably larger than near-pristine small footprint basins (Supplementary Figures 9 and 10). Small-to-medium footprint basins tend to be in somewhat warmer climates than extremely high-latitude small footprint basins and are near equilibrium until the late 19th century. In these basins, LULCC effects are small enough that they are offset by secondary forest regrowth and enhanced vegetation growth due to $CO_2$ fertilization. The most distinctive feature of small and small-to-medium footprint basins relative to the other footprint basins is the lack of any significant decrease in land N storage during the historical period.

Medium footprint basins (Fig. 6c, f; e.g., Amazon) feature significant deforestation and logging at a very early stage of the Anthropocene (Supplementary Figure 9b) followed by minimal

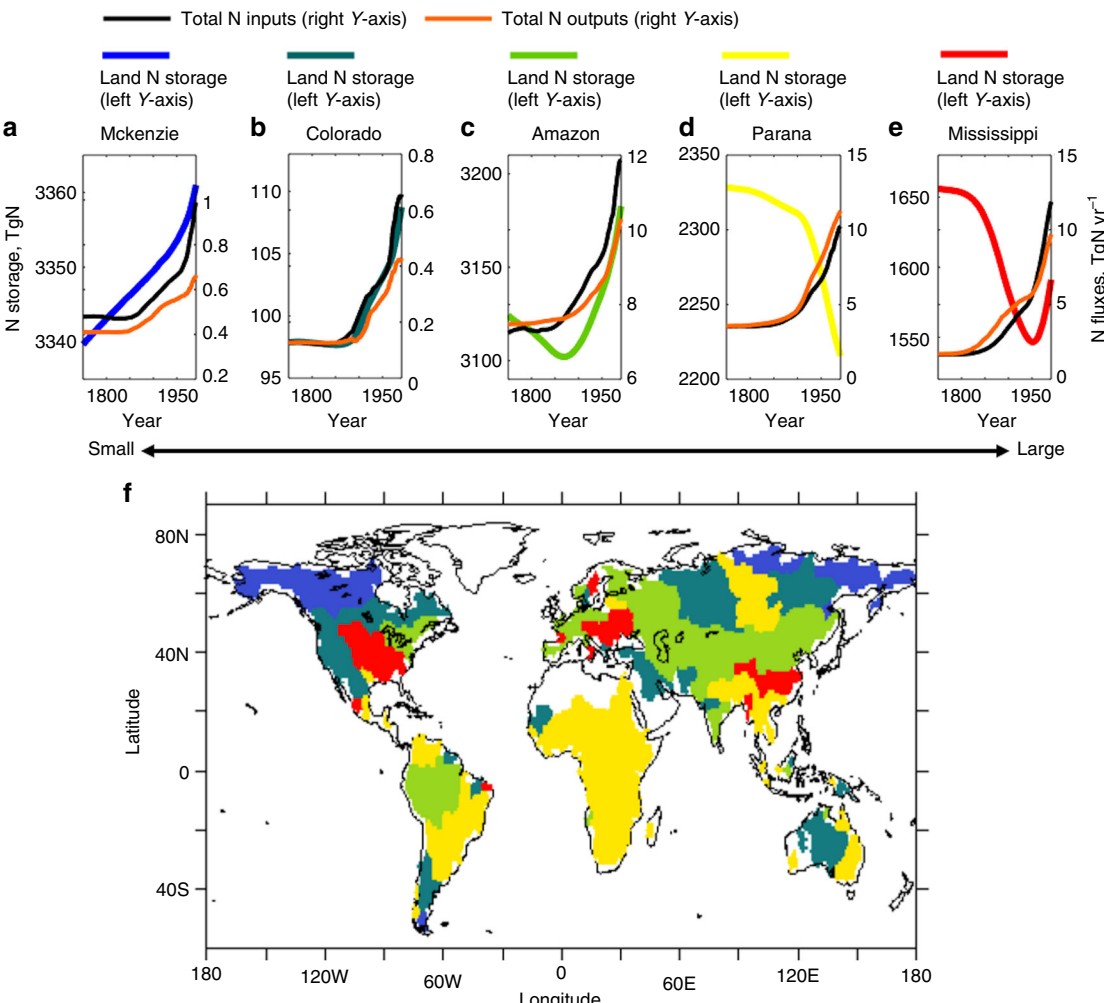

**Fig. 6** Small-to-Large LULCC footprints. **a–e** Small (blue), small-to-medium (blue-green), medium (green), medium-to-large (yellow), and large (red) LULCC footprints, reflected by five distinct basin-scale pathways of land N storage from 1750 to 2005. **f** 159 river basins shaded by five distinct colors representing the five LULCC footprints. Small footprint basins account for 21 of the 159 basins, 7% of the area of the 159 basins, and 1% of the total N outputs from the 159 basins. Small-to-medium footprint basins account for 40 basins, 21% of the area, and 10% of the outputs. Medium footprint basins account for 30 basins, 27% of the area, and 25% of the outputs. Medium-to-large footprint basins account for 57 basins, 37% of the area, and 48% of the outputs. Large footprint basins account for 11 basins, 8% of the area, and 15% of the outputs

agricultural land use over the 20th century (Supplementary Figure 9a). As a result, these basins present decreased land N storage until the late 19th century, then increasing land N storage thereafter. This pattern in other medium footprint boreal and temperate basins (Fig. 6f) are largely explained by forest management and recovery from past fire and disturbances (Europe)[43,44] or large-scale afforestation (China)[45].

Medium-to-large footprint basins (Fig. 6d, f; e.g., Parana) are characterized by early deforestation and modest agricultural land use until the late 19th century, followed by accelerating agricultural land use and large forest losses[44] (Supplementary Figure 9). This results in persistently decreasing N storage throughout the historical period, despite $CO_2$ fertilization effects since the late 19th century. It is notable that this is the only type showing continuing rapid declines in contemporary land N storage.

Finally, large footprint basins (Fig. 6e, f; e.g., Mississippi) present sharply decreased land N storage up to the mid 20th century, indicating early, rapid, and intensive LULCC[46]. These basins then exhibit a robust shift to increasing N storage due to forest expansion and recovery from past LULCC[47–49]. Increasing storage is spurred further by the effects of $CO_2$ fertilization and increasing anthropogenic Nr inputs[50].

Inspection of the basin-type distribution across latitudes shows the prevalence of basins recovering from past LULCC or, in some cases, pristine basins in temperate regions[43–45,47–49]. Tropical basins, in contrast, are characterized by many medium-to-large footprint basins featuring recently accelerating forest clearing, burning, and agricultural practices, and producing high N pollution contributions[44,51]. There are also scattered small-to-medium and medium footprint basins in the tropics, but the tempering effect of land N sequestration in these systems cannot curtail the overall prominence of the tropics in contemporary N pollution.

A breakdown of N inputs and outputs from two of the largest tropical river basins, the medium footprint Amazon and the medium-to-large footprint Parana (Fig. 7), provides further perspective on the fate and drivers of tropical N fluxes and pollution (and allows us to relate our results to recent measurements in tropical systems, see Discussion). Since the late 19th century, increasing N outputs from the both basins are primarily driven by net harvest (Fig. 7c, d) and thus result primarily in increasing atmospheric emissions. In the Parana River Basin, increasing agricultural land use, which is reflected by agricultural net harvest, is the major cause of large increases in N outputs. The increasing N outputs (~8 TgN yr$^{-1}$) are only partly balanced by increasing inputs of ~6 TgN yr$^{-1}$, driven mainly by agricultural BNF (Fig. 7a, c). Release of ~2 TgN yr$^{-1}$ of legacy land N storage thus have further augmented the outputs (Fig. 7e). In the Amazon River Basin, both non-agricultural and agricultural BNF increase inputs by ~5 TgN yr$^{-1}$ (Fig. 7b). Outputs, however, increase by only ~3 TgN yr$^{-1}$, indicating net land N sequestration of ~2 TgN yr$^{-1}$ (Fig. 7d, f). Similar to the case for the tropical systems as a whole (Fig. 5), the $CO_2$ fertilization effect leads to only a small increase in overall N outputs (Fig. 7c, d, sold vs. dashed lines). Overall BNF is higher than that in the no-$CO_2$ fertilization scenario (Fig. 7a, b). This enhances land N storage primarily in intact forests (Fig. 7e, f), but does not alter the relative magnitudes of tropical and extratropical N outputs (Fig. 7c, d).

## Discussion

Our results suggest that the tropics play a prominent role in the recent rise of global N pollution despite receiving much lower amounts of anthropogenic Nr inputs than the extratropics. This

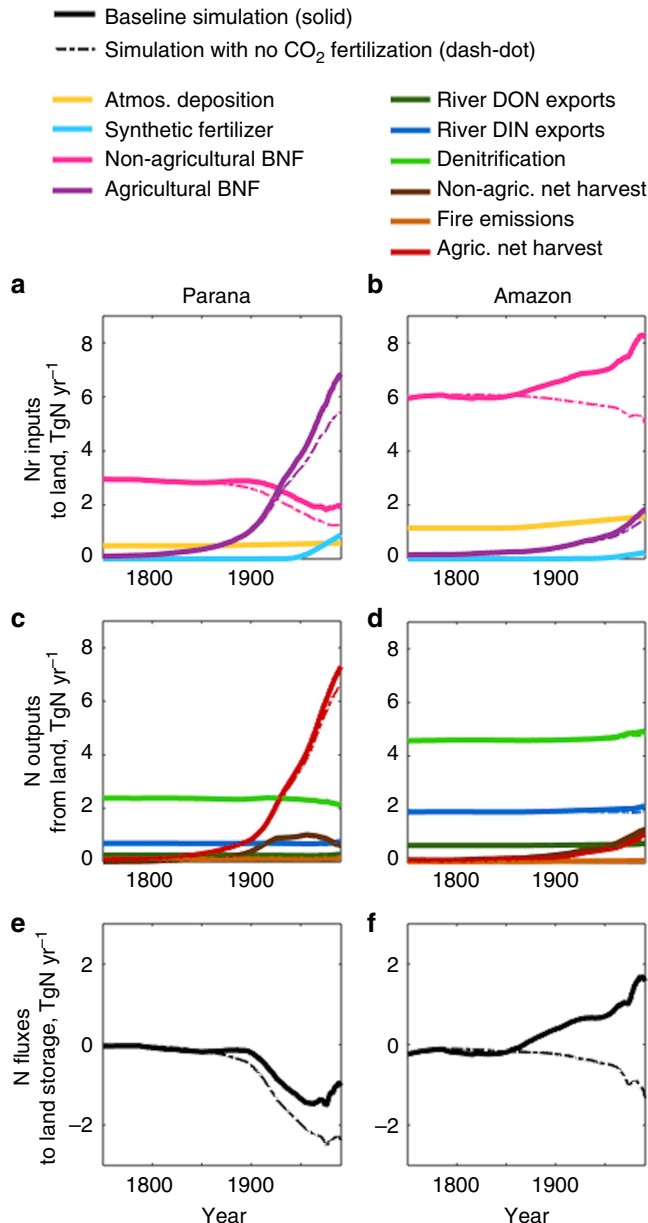

**Fig. 7** Land N fluxes in the Amazon and Parana Basins. **a, b** Land Nr inputs include atmospheric deposition (light orange), synthetic fertilizers (sky-blue), biological N fixation (BNF) in non-agricultural (plum) and agricultural (purple) lands. **c, d** Land N outputs include river dissolved organic N (DON) exports (green), river dissolved inorganic N (DIN) exports (blue), soil and freshwater denitrification (light green), fire emissions (orange), net harvest in agricultural (red) and non-agricultural (brown) lands. **e, f** N fluxes to land storage. Solid thick lines show the baseline simulation. Dash-dot thin lines show the no-$CO_2$ fertilization simulation. All plots show 30-year moving averages from 1750 to 2005

result arises due to the prevalence of tropical basins undergoing intensive LULCC since the 20th century, combined with enhanced land N sequestration in many extratropical basins recovering from past LULCC, which has slowed increases in N pollution from these regions. These results are found to be robust to the different BNF settings, fertilizer inputs, LULCC scenarios, partitioning of land N outputs into pollution, and to the exclusion of $CO_2$ fertilization.

As mentioned previously, the global transition of land from a net N source to sink in the late 1940s (Fig. 2a) is consistent with a

similar transition in terrestrial C storage[33]. Our results are also consistent with published estimates of C balance in various tropical systems (Supplementary Note 3). Tropical land as a whole being nearly N neutral over the late 20th century (Fig. 3a) aligns with filtered inverse models against an additional observational constraint, suggesting nearly neutral net C fluxes from tropical land[52]. Many plot measurements in Amazonian intact forests demonstrated long-term increases in the aboveground biomass density since 1983 (e.g., Brienen and colleagues[53]), and a similar pattern during 1983–2005 was captured in our simulation (Supplementary Figure 11). For the same period (1983–2005), however, our simulation suggests a net C source from the whole tropical forests (including both intact and disturbed forests), in a manner consistent with a recent satellite-data-based study of Baccini and colleagues[41] for 2004–2014 (Supplementary Note 3). These patterns have been attributed to a decline in forest area and forest disturbance across the tropics[41,44].

In the Parana River Basin, large increases in land N outputs (Fig. 7c) are only partially explained by increases in land Nr inputs, including agricultural BNF and fertilizer applications (Fig. 7a), leading to a Baccini-like decline in land N storage (Fig. 7e). It is the prevalence of the Parana-type basins across the tropics (Fig. 6) that underlies the idea that increasing tropical N outputs derive from release of legacy land N storage and recent accelerating agricultural land use, relative to the largest contribution of fertilizer inputs to increasing extratropical N outputs (Fig. 5b, d). In contrast, agricultural development is not nearly as widespread in the Amazon River Basin (Supplementary Figure 9, Fig. 7b, d), leading to a more Brienen-like response at the basin scale, with accumulation of land N storage (Fig. 7f).

Our results, those of Brienen and colleagues[53], and other bottom-up estimates[37] are indicative of enhanced growth and C demand in intact forests, hypothesized to result from $CO_2$ fertilization[35]. In LM3-TAN, BNF responds dynamically to enhanced C demand[28], and thus increases particularly in Amazonian non-agricultural lands including large intact forests (Fig. 7b). Although this result is plausible, it should be noted that large uncertainty remains in these simulations with little empirical evidence for BNF in intact forests increasing with $CO_2$ fertilization. While our conclusions concerning the contribution of tropical systems to global N pollution is insensitive to this response, further observational work and investigation with multi-model ensembles[54] could help resolve this uncertainty.

An important factor that is not resolved in our model is the potential role of phosphorous (P) in limiting tropical production. We would expect that BNF in response to $CO_2$ fertilization is particularly restricted in P-limited tropical intact forests[55]. We would not expect, however, that the BNF response is restricted in mostly N-limited tropical regrowing and temperate forests[56,57] or agricultural lands. Like the no-$CO_2$ fertilization case, overall N outputs would be ultimately most strongly shaped by accelerating LULCC. That is, despite the restricted BNF in tropical intact forests, overall tropical N outputs would be only modestly reduced, as the output fluxes would be supplied through release of legacy land N storage. Indeed, the P-limited tropics combined with enhanced land N sequestration in the N-limited and $CO_2$ fertilized extratropics would likely accentuate the prominence of the tropics as a source of global N pollution.

The lower tropical land capacity to sequester Nr inputs will challenge efforts to control atmospheric and oceanic pollution in tropical regions, and control tropical contributions to global N pollution. Tropical pollutant loadings to the ocean and atmosphere are expected to further grow with dramatic increases in Nr inputs and LULCC in tropical regions over the next few decades[1,22,24,25]. The resulting high N fluxes will exacerbate hypoxia and phytoplankton blooms in tropical coastal waters[58]

and amplify nitrous oxide emissions from tropical land[51,59]. It is thus critical that tropical Nr inputs, LULCC, and legacy N are thoroughly considered in future coastal eutrophication and greenhouse gas emission policies and mitigation strategies.

## Methods

**Model description**. Geophysical Fluid Dynamics Laboratory (GFDL) Land Model LM3-Terrestrial and Aquatic Nitrogen (TAN) was developed by incorporating a global river routing and lake model[29], river N cycling processes, and new terrestrial N cycling processes and inputs, such as soil denitrification and point N sources to rivers (i.e., urban wastewater discharges)[27], into the coupled C and N (C-N) Terrestrial Ecosystem Model (TEM) LM3V-N[28,30]. TEMs[28,60] capture coupled terrestrial C-N dynamics that critically affect the state of N storage in vegetation and soils, such as deforestation for agriculture, wood harvesting, and reforestation after harvesting. TEMs extended to include global river routing models[27,61] are thus well suited to simulate hydrological N leaching from soils to rivers, and track the ultimate fate of land N to the ocean and atmosphere. While measurement-based N constraints on TEMs rely on global terrestrial N budgets that are known to include substantial uncertainty[1,22,23], LM3-TAN can be also constrained with relatively abundant measurements of regional N loads and concentrations from globally-distributed rivers, spanning various climates, biomes, and human activities (Supplementary Figures 1 and 2, Supplementary Table 2).

Within LM3-TAN, water and energy storage in land and exchanges with the atmosphere are computed at a 30-min timestep. Five vegetation functional types (C3 and C4 grasses, temperate deciduous, tropical, and cold evergreen trees) are simulated based on total biomass and prevailing climate conditions[30]. The vegetation pools include leaves, fine roots, sapwood, heartwood, and labile storage, which are updated daily to account for vegetation growth and allocation, leaf fall and display, natural and fire-induced mortality. Four land-use types (primary land—land undisturbed by human activities during land-use reconstruction, secondary land—abandoned agricultural land or regrowing forest after logging, cropland, and pasture) and land-use changes are simulated by using a historically reconstructed scenario of land-use transitions including the effects of wood harvesting and shifting cultivation[15]. The model is spatially gridded, and each grid cell consists of up to 15 tiles: 1 primary land, 1 cropland, 1 pasture, 1 lake, 1 glacier, and up to 10 secondary land tiles, reflecting unique perturbation histories, such as de/reforestation and agricultural activities.

When wood is harvested, the vegetation biomass in primary or secondary land is removed, and the land is replaced with (1) cropland or pasture for agriculture or (2) secondary forests after logging[30]. When crops are harvested, all leaf and the aboveground fraction of labile stores in croplands are cut. When grass is harvested, a fraction of leaf in pasture is removed. Crop and grass harvesting are done annually. See Shevliakova and colleagues[30] for a detailed harvesting description. As described in the main text, net harvest is defined as harvest minus manure applications[8] and urban wastewaters[31], and the majority of the net harvest is assumed to be released to the atmosphere[1,6,8] (Supplementary Table 1, Supplementary Note 1). The net harvest can be further partitioned into that associated with agricultural lands (agricultural net harvest) and that associated with non-agricultural lands (non-agricultural net harvest). Agricultural net harvest is estimated as the sum of harvest in cropland and pasture tiles minus manure applied to croplands[8]. Non-agricultural net harvest is the sum of harvest in all the other tiles minus urban wastewater discharges[31].

LM3-TAN has 4 organic soil pools (fast/slow litter and slow/passive soil) and 2 inorganic soil pools (ammonium and nitrate plus nitrite)[28]. Each C pool in vegetation and organic soils is paired with a respective N pool using pool-specific C:N ratios. Decomposition of soil organic matter (SOM) releases inorganic ammonium N, which is further transformed into nitrite or nitrate N by nitrification. Soil denitrification emissions are simulated by using a first-order loss function with respect to soil nitrate N content, with adjustments for the influence of soil water content and temperature[27]. Simulations of biologically available inorganic N species allow accounting for N limitation on vegetation growth and BNF as well as N feedbacks on SOM decomposition and immobilization. BNF is considered as a high light-requiring costly process, and is thus only simulated if plant uptake cannot meet N demands, as a prognostic function of local N availability and of sunlight access in temperate and boreal biomes[28]. Simulated BNF can be further partitioned into that associated with agricultural lands (agricultural BNF) and that associated with non-agricultural lands (non-agricultural BNF). Agricultural BNF is estimated as the sum of BNF in cropland and pasture tiles and non-agricultural BNF is the sum of BNF in all the other tiles.

Rivers receive dissolved organic, ammonium, and nitrate plus nitrite N via hydrological soil leaching[27] and urban wastewater discharges[31]. River N is routed downstream with river flows. Microbial processes of mineralization, nitrification, and denitrification in rivers are simulated by using first-order loss function with respect to N content and with an adjustment for the influence of water temperature. See Lee and colleagues[27] for a detailed model description and structure. Lentic systems (e.g., lakes and reservoirs) have long residence times relative to rivers, providing an extended opportunity for N processing[62]. Despite this, previous TEMs including global river routing models[27,61] did not account for lentic N dynamics. We thus extended LM3-TAN by incorporating a simple model

of dynamic lake N cycling processes (Supplementary Note 4). The resultant model couples hydrological, ecological, and biogeochemical cycles, and captures key N dynamics within a vegetation-soil-river-lake system in response to LULCC, anthropogenic Nr inputs, atmospheric $CO_2$, climate change and variability. Newly introduced or adjusted parameters from the earlier developments are summarized in Supplementary Table 3.

**Model forcing and simulations.** Following ~11,000 years of spin-up, LM3-TAN simulated global land N storage and fluxes to the ocean and atmosphere from 1700 to 2005 with historical forcings of atmospheric $CO_2$, anthropogenic Nr inputs[8,9,31], climates[32], and LULCC[15] at $1 \times 1$ degree resolution. Before 1700, the land surface was assumed to be undisturbed by human activities. LULCC were simulated from 1700 to 2005[15]. An uncertainty test was also conducted using an alternative LULCC scenario that does not account for shifting cultivation[15]. The model was forced by observation-based, global near-surface meteorology data, including precipitation, specific humidity, air temperature, surface pressure, wind speed, and short- and long-wave downward radiation at 3-h and 1-degree resolution[32]. The forcing data were cycled over a period of 30 years (1948–1977) to perform long-term simulations up to 1947, and the 1948–2005 forcing data were used for the simulations from 1948 to 2005. Preindustrial $CO_2$ concentration of 286 ppmv was applied up to 1800, and reported annual $CO_2$ concentrations from the NOAA Earth System Research Laboratory (available at [http://www.esrl.noaa.gov/gmd/ccgg/trends/global.html]; last access: 25 February 2019) were applied from 1801 to 2005.

Global Nr inputs to the land biosphere include atmospheric deposition for two representative years (1860 and 1993)[9], synthetic fertilizers and livestock manure for four years (1900, 1950, 2000, and 2050)[8], and urban wastewater discharges for four years (1970, 1990, 2000, and 2030)[31]. We compiled the country-specific urban wastewater discharges and administrative population data from the Environmental Systems Research Institute, Inc. (World Administrative Units 2002, available at [http://map.princeton.edu/search]; last access: 19 May 2018) to produce sub-country level global urban wastewater Nr inputs. The four Nr inputs were linearly interpolated between years of data available to be applied for the intervals without data. Uniform annual rates of the first (last) year data were applied for the periods before (after) the first (last) year of data available. Atmospheric deposition was applied for the entire simulation periods. Fertilizers and manure were applied from 1901 to 2005. Urban wastewaters were applied from 1951 to 2005. Given high uncertainty in Nr inputs as discussed in Supplementary Table 1 and Supplementary Note 1, we also ran simulations with a range of BNF settings[1,9] and different fertilizers[40].

Atmospheric deposition was applied to all land-use tiles. Fertilizers allocated to three crop groups (i.e., wetland rice, leguminous crops, and other upland crops) of Bouwman and colleagues[8] were applied to cropland tiles. Fertilizers allocated to grassland in mixed systems (MG) of Bouwman and colleagues[8] were applied to pasture tiles. Stored or collected manure for applications to cropland in mixed systems (MC) and areas dominated by pastoral grazing (PC) of Bouwman and colleagues[8] was applied to cropland tiles. This recycled manure to fertilize crop production was deducted from simulated harvest. Urban wastewaters were directly applied to rivers. These urban wastewaters were also deducted from simulated harvest. Each of the Nr inputs was divided into three N species (i.e., organic, ammonium, and nitrate plus nitrite N) by multiplying reported fractions (Supplementary Table 4)[63,64]. Sensitivity tests show that changing fractionation of N species has almost no influence on global, tropical, and extratropical land N fluxes (Supplementary Figure 12). Each of the N species for the four Nr inputs was applied to the corresponding soil and river pools respectively[27].

By the end of the spin-up, the N budget has come to equilibrium for all regions, except high-latitude boreal forest and tundra regions where N continues to accumulate in cold soils. We note that some degree of disequilibrium in such high-latitude systems may reflect the actual multimillennial response to the Last Glacial Maximum. This response has previously been modeled in other long-term N simulations[42]. These systems account for only 1% of the global land N outputs.

**Land N pollution and uncertainty analyses.** The input and output fluxes described in the main text and Methods allow us to close the land N budget and assess whether land is sequestering or releasing N at global and basin scales. Estimating N pollution from land, however, requires additional considerations. Total land N pollution was estimated as the total land N outputs, minus the sum of $N_2$ emissions and human appropriation of the net harvest into durable goods (e.g., home building). The fractional partitioning of each output flux into these environmentally benign forms was based on reported values from the literature[1,6,51,60,65–67] (Supplementary Note 2). A Monte Carlo randomization across the uncertainty in these fractions was used to estimate global and tropical land N pollution and uncertainty in these estimates.

We define a baseline simulation which simulated global BNF near the center of published ranges[1,9] and used fertilizer inputs from Bouwman and colleagues[8] and a LULCC scenario including the effects of wood harvesting and shifting cultivation[15]. To further quantify uncertainties, we repeated simulations for BNF settings spanning the upper and lower bounds of the published ranges[1,9], different fertilizer inputs[40], and a different LULCC scenario without shifting cultivation[15]. Since organic forms of N are generally less immediately bioavailable than inorganic N[68],

organic N exports from rivers may be less critical drivers of acute pollutant effects on coastal ecosystems. We thus tested the robustness of our results to the exclusion of river organic N exports from land N pollution. Our reported uncertainties consider the sensitivity of our results to variations in this forcing. Finally, in recognition of the uncertainty surrounding the degree of $CO_2$ fertilization effects on terrestrial C sink[69,70], we also consider the robustness of our results to a scenario with no $CO_2$ fertilization.

**LULCC footprints.** We classified the 159 river basins based on five small-to-large evolving types of LULCC footprints, which are reflected by five distinct basin-scale pathways of land N storage from 1750 to 2005: Small footprint (persistently increased N storage; Fig. 6a, f, blue), Small-to-medium footprint (stable and increased N storage; Fig. 6b, f, blue-green), Medium footprint (decreased and increased N storage; Fig. 6c, f, green), Medium-to-large footprint (increasingly reduced N storage; Fig. 6d, f, yellow), and Large footprint (sharply reduced and increased N storage; Fig. 6e, f, red). The land N storage from 1750 to 2005 in the Mackenzie, Colorado, Amazon, Parana, and Mississippi River Basins were chosen as representatives of reflecting the five LULCC footprints, because these basins are the largest ones presenting each type among the 47 basins used for the result evaluation (Supplementary Table 2). This classification was made based on the highest Pearson's correlations between each of the land N storage in the 154 basins and the five representatives from 1750 to 2005. The result was shaded in five distinct colors, representing each LULCC footprint (Fig. 6f).

## Data availability
The authors declare that LM3-TAN simulation results are available from the corresponding author upon request. All other data supporting the findings of this study are available within the paper.

## Code availability
The authors declare that LM3-TAN codes are available from the corresponding author upon request.

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

## Acknowledgements

We thank John Dunne from NOAA/Geophysical Fluid Dynamic Laboratory and two anonymous reviewers for their incisive comments on the manuscript. M.L. was supported by the NOAA marine ecosystem tipping points initiative and the NOAA (U.S. Department of Commerce) grant NA14OAR4320106.

## Author contributions

M.L., E.S., and C.A.S. designed the research. All authors contributed to develop the model and its global implementation. M.L. performed the model simulations and analyses. All authors analyzed and discussed the results. M.L., E.S., and C.A.S. wrote major portions of the manuscript with substantial input from S.M. and P.C.D.M.

## Additional information

**Competing interests:** The authors declare no competing interests.

