## [Peer Review File · Nature Communications]

Reviewers' comments:

Reviewer #1 (Remarks to the Author):

Lee et al. present an interesting model analysis of the global N cycle and its response to global change drivers since 1700. They use a modified version of LM3VN that incorporates river and lake N processing as well as new sensitivity analyses for several parameters including BNF. They address an important set of questions and draw quite novel and provocative conclusions—namely that the terrestrial biosphere has switched from net N source to sink but that tropical basins are increasingly inefficient at retaining N relative to reactive N (Nr) inputs. This represents a ton of work and I commend the authors on the effort. I really like the comparison of model output to observed river N loads.

While I generally enjoyed reading the paper (many times) and found aspects new and interesting, I struggle with the main inference drawn (increasing ocean and atmospheric N pollution from the tropics) as reflected in the title itself. Based on the model description, data sources and uncertainty analyses described in the main text and the supplemental, this conclusion is just a little hard to buy. Perhaps, it is actually a robust finding, but as written, many aspects of the analyses would require further explicit clarification to convincingly demonstrate this. The idea that increased tropical N losses derive from legacy N due to increasing LULCC is not strongly supported. How does this finding align with observed net aboveground C sink in tropical forests (Brienen et al. Nature 2015) versus tropical forests overall being a net C source due to LULCC (Baccini et al. Nature 2018)? Perhaps presenting an explicit regional analyses of Nr inputs and losses would help clarify. For example, what LULCC is going on in the Parana basin that results in such high N losses? Deforestation, conversion to soybean? In the case of the Amazon, the estimated contemporary river N flux is very similar to previous estimates for pre-industrial fluxes suggesting that the increased losses are gaseous. However, it is not entirely clear to me how gas losses were partitioned into N₂O versus N₂ or how total dissolved N losses were partitioned into DIN and DON. These have huge implications for N limitation and export. N₂ is likely the dominant gas loss from intact lands and N₂ is obviously not a pollutant. Much of DON is thought to be less bioavailable and while some can be mineralized in estuaries and oceans, it is not clear that this should be expected to increase over time and represents a source of “pollution”.

At steady state, N saturation can be quickly diagnosed by DIN losses > atmospheric deposition or if DON losses < BNF (Menge. Ecosystems. 2011). The degree to which denitrification represents a demand-independent loss is not resolved but has large implications for N limitation of plants especially in the tropics where natural N₂ losses likely dominate (Brookshire et al. GRL. 2017). These issues are relevant to your proposed NIL but it is not clear on exactly how Nr inputs are defined. Is this just atmospheric N dep and fertilizer and manure? If so, OK. But if BNF is included, that is not correct. Further, while there is plenty of evidence that massive LULCC in some tropical areas can

increase river N, DIN and N₂O losses are often lower in secondary aggrading forests and increase as N accumulates (Davidson et al. Nature 2007). I can believe the increase in tropical N losses if these derive primarily from agriculture (Tian et al. Nature 2016). As far as BNF goes, I appreciate the authors attempt to address uncertainty. It is difficult to measure and scale in the field and it is poorly simulated in most models. LM3VN may very well be the best with BNF and I understand how, because it responds dynamically and is demand driven, BNF increases over time in simulations, there is virtually no evidence for this in nature. Only a handful of FACE sites ever observed any change in BNF. While there is good evidence that BNF is higher in aggrading secondary forest (and thus with increasing LULCC one could argue this should increase tropical BNF inputs) it is not clear that this is what is driving the simulated tropical BNF increase versus CO₂. I think it highly implausible that tropical BNF has doubled since 1700. Another source of confusion is how “harvest emissions” are defined and modeled. Maybe I’m really missing something but I read the methods many times and still don’t quite get it.

Next, the writing and English need significant work throughout. Examples are included in specific comments below.

Finally, there is a wealth of references on tropical N cycling, limitation and efficiency that are ignored. Some of the citations provided are rather strange and not well supported.

Page1, Line 23: “Grown anthropogenic Nr...”?

P1,L27: 30% more is astonishing but if this is relative to dep and fert inputs only and includes all N losses (DIN, DON, N₂, N₂O) this doesn’t seem like a fair comparison per comments above.

P1, L42: “rapidly grown”?

P1,L47: Again, there is very little empirical evidence for BNF increasing with CO₂. It makes sense theoretically as BNF increases to meet CO₂ induced demand but just not supported by evidence. Also this citation (12) doesn’t work as they don’t really even address BNF.

P2, L6: True, that field description of many N cycling processes are less frequent in the tropics especially BNF and N₂ losses. However, there are hundreds of empirical studies on many dimensions of tropical N cycling.

P2,L8: What is this saying? Of course natural BNF has added significant N since preindustrial times, but there is little evidence that it has increased.

P2, L27: "emission associated with harvest"...What? Also the English doesn't work. "Harvesting"? are these crops?

P2,L33: This makes little sense. If the N₂ losses are derived from haber-Bosch then yes, I see your point. However, most N₂ losses from tropics for example likely do not but are natural.

P3, L15: This index needs to be reevaluated. Given historically low dep and fert inputs into many tropical regions, of course the index is going to be sensitive given that all forms are included in losses while only N_r is included in inputs.

P3, L23: What does this mean?

P3, L35: "63% of global harvest emissions". Is this deforestation? What is this?

Supplemental Page 1, L17: Vitousek actually reports much lower isotope based estimates for BNF (58 Tg N yr⁻¹)

S1,L21: Huh? No influence of different BNF levels do not have much influence? I get that is insensitive to low versus high but it is unclear how the literature range was incorporated in the simulations. Are these used to constrain preindustrial spin up or to match contemporary observations. The references used represent a mixture of preindustrial BNF estimates and contemporary. It is also worth looking into Cleveland et al (PNAS 2013) for satellite and modeling based estimates of contemporary biome BNF rates for unmanaged lands.

S1, L26: Confusing

S1, L34: Again "harvest" needs more explanation

S1,L39: How N₂ was modeled needs much more clarification. For tropical forest denitrification (total N gas emissions) estimates see Bai et al. (Biogeosciences 2012) and Brookshire et al. (GRL 2017)

Reviewer #2 (Remarks to the Author):

This is a revised version of manuscript I reviewed before, which was submitted to Nature. Lee et al applied the GFDL Land model -Terrestrial and Aquatic Nitrogen (LM3-TAN) to analyze the effect of the past two and half centuries of anthropogenic N_r inputs, land-use and land-cover changes (LULCC), increasing atmospheric CO₂, and climate on global and basin-scale N fluxes to the oceans and atmosphere. Major conclusions include:1) Globally, land transitioned from a net N source to N sink in the late 1940s, largely due to enhanced vegetation growth under elevated atmospheric CO₂; 2) Tropical land has produced more than 50% of contemporary land N outputs despite covering only 34% of global land area and receiving far lower synthetic fertilizer applications than the extratropics. Expanding LULCC in tropical regions could shift the global land balance back to a net N source. This revised manuscript has addressed many of my concerns. Below are some comments needed to be addressed before publication.

1) As the authors stated, the factor driving land from N source into N sink is elevated CO₂ concentration. The sensitivity analysis of removing CO₂ fertilization effect delays such a global land transition from a N source to N sink (Fig 2b). Thus, the accurate simulation of nutrients limits to the CO₂ effect on vegetation growth and the consequent biomass accumulation is critical. Large uncertainty exists in CO₂ fertilization effect on vegetation growth, particularly there is lack of observational evidence to quantify the magnitude of CO₂ fertilization effect on vegetation growth and BNF. It's necessary to have a discussion on this uncertainty in the manuscript.

2) It has been long recognized that tropical forests' growth are largely limited by phosphorus (P) availability. I noted that TEM -C-N model used in LM3_TAN does not account for P limits, which can potentially overestimate CO₂ impacts on C and N cycling. A discussion on this uncertainty is also needed.

3) Some key indicators such as NLI need statistical test for significance.

4) If you have implemented a set of simulations with different input data such as BNF settings, different fertilizers, different LULCC scenarios, etc, I suggest you report your results as mean \pm deviation.

5) A discussion on uncertainty analysis should be a necessary component of this work, which should be added to the main body of this manuscript.

Reviewer #1 (Remarks to the Author):

Lee et al. present an interesting model analysis of the global N cycle and its response to global change drivers since 1700. They use a modified version of LM3VN that incorporates river and lake N processing as well as new sensitivity analyses for several parameters including BNF. They address an important set of questions and draw quite novel and provocative conclusions—namely that the terrestrial biosphere has switched from net N source to sink but that tropical basins are increasingly inefficient at retaining N relative to reactive N (N_r) inputs. This represents a ton of work and I commend the authors on the effort. I really like the comparison of model output to observed river N loads.

Response 1: We'd like to thank the reviewer for their positive comments and encouragement.

While I generally enjoyed reading the paper (many times) and found aspects new and interesting, I struggle with the main inference drawn (increasing ocean and atmospheric N pollution from the tropics) as reflected in the title itself. Based on the model description, data sources and uncertainty analyses described in the main text and the supplemental, this conclusion is just a little hard to buy. Perhaps, it is actually a robust finding, but as written, many aspects of the analyses would require further explicit clarification to convincingly demonstrate this.

Response 2: We're sorry to hear that our submitted draft did not fully convince the reviewer regarding the robustness of our primary conclusion – the prominence of the tropics as a source of global N pollution. We have extended our analyses and made substantial revisions to address the reviewer's specific concerns (described in detail below). These changes more convincingly demonstrate and communicate the robustness of our results, and we would like to thank the reviewer for pushing us on these points.

The idea that increased tropical N losses derive from legacy N due to increasing LULCC is not strongly supported. How does this finding align with observed net aboveground C sink in tropical forests (Brienen et al. Nature 2015) versus tropical forests overall being a net C source due to LULCC (Baccini et al. Nature 2018)? Perhaps presenting an explicit regional analyses of N_r inputs and losses would help clarify. For example, what LULCC is going on in the Parana basin that results in such high N losses? Deforestation, conversion to soybean?

Response 3: Our results are consistent with both Brienen et al. (2015) and Baccini et al. (2017). Brienen et al. (2015) demonstrated a long-term increase in the aboveground biomass density in Amazonian intact forests since 1983. A similar pattern during 1983-2005 was captured in our simulations (See Supplementary Fig. 11 for a direct comparison). However, our analysis of aboveground C changes in entire tropical forests (including both intact and disturbed forests)

suggests a net C source for the same period (1983-2005), in a manner consistent with Baccini et al. (2017) (Supplementary Note 3). These patterns have been attributed to a decline in forest area and forest disturbance during LULUC across the tropics (Baccini et al., 2017; Hansen et al., 2013). We have added discussion of our results relative to Brienen et al. (2015) and Baccini et al. (2017) on Page 6, Lines 31-44.

Response 4: As the reviewer suggested, we now present an explicit regional analysis of N inputs, outputs, and fluxes to land storage in the Parana and Amazon River Basins (Fig. 7) on Page 6, Lines 4-19. In the Parana River Basin, recent accelerating agricultural land use and transition of natural to agricultural lands (i.e., LULUC) (Supplementary Fig. 9) lead to large increases in land N outputs (Fig. 7c). The increasing N outputs are only partially explained by increasing Nr inputs, including agricultural BNF and fertilizer applications (Fig. 7a), leading to a Baccini-like decline in land N storage (Fig. 7e). Agricultural development is not nearly as widespread in the Amazon River Basin (Supplementary Fig. 9, Fig. 7b and d), leading to a more Brienen-like response, with accumulation of land N storage (Fig. 7f). The relevant text discussing these responses is now on Page 6, Lines 46-51 to Page 7, Lines 1-2.

Response 5: It is the prevalence of the Parana-type basins across the tropics (Fig. 6) that underlies the idea that increased tropical N losses derive from release of legacy land N storage and recent accelerating agricultural land use, relative to the largest contribution of fertilizer inputs to extratropical N losses (Page 6, Lines 46-51 to Page 7, Lines 1-2). The robustness of our primary findings, including the prominence of the tropics as a source of global N pollution, was tested over a range of BNF settings, different fertilizer inputs, different LULUC scenarios, scenarios with and without CO₂ fertilization, and the partitioning of N outputs into environmentally benign vs pollutant forms. We now more fully and consistently describe these baseline and uncertainty simulations (Page 3, Lines 6-16) and present these uncertainties throughout the manuscript (Fig. 3 and 4, Supplementary Fig. 3-8).

In the case of the Amazon, the estimated contemporary river N flux is very similar to previous estimates for pre-industrial fluxes suggesting that the increased losses are gaseous.

Response 6: Yes, the added regional analysis of the Amazon River Basin (Fig. 7b, d, f) demonstrates very stable river N fluxes to the ocean, yet notably increased net harvest – N in harvested woods, crops, and grasses after subtracting out manure applied to croplands (Bouman et al., 2013a) and urban wastewater discharges (Van Dreht et al., 2009). The majority of the net harvest is assumed to ultimately reach the atmosphere via various pathways including wood, biofuel, and waste burning, livestock respiration, emissions from food, human, and livestock (Galloway et al., 2004; Ciais et al., 2013; Bouwman et al., 2013). This result thus implies that the increased N outputs from the Amazon River Basin are mostly gaseous (Page 6, Lines 4-9). We

have expanded descriptions of the net harvest in much detail in main text (Page 2, Lines 39-46), Methods (Page 8, Lines 2-11), and Supplementary Information (Supplementary Note 1 and 2).

However, it is not entirely clear to me how gas losses were partitioned into N₂O versus N₂ or how total dissolved N losses were partitioned into DIN and DON. These have huge implications for N limitation and export. N₂ is likely the dominant gas loss from intact lands and N₂ is obviously not a pollutant. Much of DON is thought to be less bioavailable and while some can be mineralized in estuaries and oceans, it is not clear that this should be expected to increase over time and represents a source of “pollution”.

Response 7: The reviewer is right to point out that the previous manuscript focused on total N outputs without sufficient analysis of the forms of N outputs. We have now significantly expanded our description and analysis of the sensitivity of our results to the partitioning land N outputs into environmentally benign vs. pollutant forms, and estimated both land N outputs and pollution to the ocean and atmosphere.

Total land N pollution was estimated as the total land N outputs, minus the sum of N₂ emissions and human appropriation of the net harvest into durable goods (e.g., home building). The fractional partitioning of each output flux into these environmentally benign forms was based on reported values from the literature (Galloway et al., 2004; Ciais et al., 2013; Bai et al., 2012; Bouwman et al., 2013b; Bouwman et al., 1993; Tian et al., 2016; Zaehle et al., 2013). A Monte Carlo randomization across the uncertainty in these fractions was used to estimate global and tropical land N pollution and uncertainty in these estimates (Page 2, Lines 48-51 to Page 3, Lines 1-4, Supplementary Note 2).

LM3-TAN explicitly simulates transformation and transport of DIN and DON in rivers and lakes and their exports to the ocean, which are described in Methods (Page 8, Lines 26-37) and Supplementary Information (Supplementary Note 4, Supplementary Table 3). The general fidelity of this partitioning with measurement-based estimates is provided in Supplementary Fig. 2 and discussed in Supplementary Note 1. To estimate the partitioning of other land N outputs, we needed three additional partitions of the fluxes directly simulated by LM3-TAN: 1) the partitioning of soil and freshwater denitrification into N₂O and N₂ emissions, 2) the partitioning of the net harvest into N₂ emissions, and 3) the partitioning of the net harvest transformed into durable goods. To test the robustness of our results to uncertainty in these partitions, an interval for each partition was assigned based on the scientific literature (See Supplementary Note 2) and 1000 Monte Carlo style calculations were conducted with random draws from a uniform distribution across the uncertainty interval. This was done for the baseline simulation, and for the 4 sensitivity simulations with the different BNF, fertilizer inputs, and LULUC scenarios. Lastly, we created additional 1000 different total land N pollution estimates by excluding river organic N exports. These provided a total of 6000 permutations.

The result of this analysis (now shown in Fig. 4) suggests that the tropics contribute $56\pm 6\%$ of global land N pollution to the ocean and atmosphere, despite covering only 34% of global land area and receiving much lower amounts of synthetic fertilizers than the extratropics (Fig. 5a, b). We even find that the conclusions are robust to the exclusion of CO₂ fertilization effects (Supplementary Fig. 4). The relevant text describing these results is now on Page 4, Lines 11-16.

At steady state, N saturation can be quickly diagnosed by DIN losses > atmospheric deposition or if DON losses < BNF (Menge. Ecosystems. 2011). The degree to which denitrification represents a demand-independent loss is not resolved but has large implications for N limitation of plants especially in the tropics where natural N₂ losses likely dominate (Brookshire et al. GRL. 2017). These issues are relevant to your proposed NIL but it is not clear on exactly how Nr inputs are defined. Is this just atmospheric N dep and fertilizer and manure? If so, OK. But if BNF is included, that is not correct.

Response 8: The NLI is a simple but informative budget diagnostic that compares total N outputs from the system against total N inputs to the system, as a ratio. It is not intended as a metric of saturation or limitation. Even unsaturated systems can release more N than they receive due to LULUC. As such, it is both correct and necessary from a budgetary perspective to include BNF as an input. We have now more clearly described all the N input and output terms (Page 2, Lines 31-46) and the definition and intended use of NLI (Page 3, Lines 48-51 to Page 4, Lines 1-5).

We also note that our NLI is estimated by considering N fluxes in all kind of land use and land cover, including intact/disturbed forests, agricultural and other lands, and freshwaters, which have not been at a steady state due to LULUC, as reflected by global and basin-scale pathways of land N storage from 1750 to 2005 (Fig. 2 and 6). Whereas, Menge (2011)'s diagnosis of whether systems are saturated with N is only applicable to the systems presumably at a steady state, such as intact forests.

Further, while there is plenty of evidence that massive LULCC in some tropical areas can increase river N, DIN and N₂O losses are often lower in secondary aggrading forests and increase as N accumulates (Davidson et al. Nature 2007). I can believe the increase in tropical N losses if these derive primarily from agriculture (Tian et al. Nature 2016).

Response 9: Yes, the increases in tropical N outputs derive primarily from agricultural land use and deforestation (i.e., LULUC). Averaged across all tropical systems, tropical land is nearly N neutral (Fig. 5e), as the increasing outputs (Fig. 5c) are in near balance with increasing agricultural BNF (largely due to expansion of agricultural land areas) and, more recently, limited increases in fertilizer inputs and atmospheric deposition (Fig. 5a) (Page 4, Lines 18-23). However, we note that there is heterogeneity between tropical basins, as discussed earlier in

Response 4 about the Parana and Amazon River Basins (Fig. 7). Especially, many tropical basins are characterized by the Parana type featuring release of legacy land N storage that has further augmented increasing N outputs (Fig. 6d, f) (Page 5, Lines 35-40 and Page 6, Lines 9-12).

As far as BNF goes, I appreciate the authors attempt to address uncertainty. It is difficult to measure and scale in the field and it is poorly simulated in most models. LM3VN may very well be the best with BNF and I understand how, because it responds dynamically and is demand driven, BNF increases over time in simulations, there is virtually no evidence for this in nature. Only a handful of FACE sites ever observed any change in BNF. While there is good evidence that BNF is higher in aggrading secondary forest (and thus with increasing LULCC one could argue this should increase tropical BNF inputs) it is not clear that this is what is driving the simulated tropical BNF increase versus CO₂. I think it highly implausible that tropical BNF has doubled since 1700.

Response 10: In Fig. 5a, we have now further partitioned BNF into that associated with agricultural lands (agricultural BNF) and that associated with non-agricultural lands (non-agricultural BNF). Our results are consistent with the reviewer's comment in that tropical BNF in non-agricultural lands, including intact and secondary forests, remains stable over that last two and half centuries. This is the result of counteracting influences of declining non-agricultural land areas and CO₂ fertilization. The tropical BNF increase is driven primarily by agriculture. We furthermore note that simulated agricultural BNF (69 (59-85) TgN yr⁻¹) is consistent with the latest and most comprehensive BNF estimate in agricultural systems (50-70 TgN yr⁻¹) (Herridge et al., 2008). This result is discussed in detail in Supplementary Note 1.

Another source of confusion is how "harvest emissions" are defined and modeled. Maybe I'm really missing something but I read the methods many times and still don't quite get it.

Response 11: We have clarified this on main text (Page 2, Lines 39-46), Methods (Page 8, Lines 2-11), and Supplementary Information (Supplementary Note 1 and 2), some of which we have pasted below for reference:

Main text (Page 2, Lines 39-46): The net harvest includes N in harvested woods, crops, and grasses³⁰ after subtracting out manure applied to croplands⁸ and urban wastewater discharges³¹, and can be further partitioned into agricultural and non-agricultural net harvest. The majority of the net harvest is assumed to ultimately go into the atmosphere via various pathways including wood, biofuel, and waste burning, livestock respiration, emissions from food, human, and livestock waste^{1,6,8}, though some is sequestered in durable goods (i.e., home building). See Methods for a detailed description of the input and output terms and Supplementary Note 1 and Supplementary Table 1 for further discussion.

Methods (Page 8, Lines 2-11): When wood is harvested, the vegetation biomass in primary or secondary land is removed, and the land is replaced with 1) cropland or pasture for agriculture or 2) secondary forests after logging³⁰. When crops are harvested, all leaf and the aboveground fraction of labile stores in croplands are cut. When grass is harvested, a fraction of leaf in pasture is removed. Crop and grass harvesting are done annually. See Shevliakova and colleagues³⁰ for a detailed harvesting description. As described in the main text, net harvest is defined as harvest minus manure applications⁸ and urban wastewaters³¹, and the majority of the net harvest is assumed to be released to the atmosphere^{1,6,8} (Supplementary Note 1, Supplementary Table 1). Agricultural net harvest is estimated as the sum of harvest in cropland and pasture tiles minus manure applied to croplands⁸. Non-agricultural net harvest is the sum of harvest in all the other tiles minus urban wastewater discharges³¹.

Next, the writing and English need significant work throughout. Examples are included in specific comments below.

Response 12: Throughout the manuscript, we have attempted to improve writing and English.

Finally, there is a wealth of references on tropical N cycling, limitation and efficiency that are ignored. Some of the citations provided are rather strange and not well supported.

Response 13: We corrected and added references as requested by the reviewer.

Page1, Line 23: “Grown anthropogenic Nr...”?

Response 14: We modified “grown anthropogenic Nr” to “increased anthropogenic Nr” (Page 1, Line 25).

P1,L27: 30% more is astonishing but if this is relative to dep and fert inputs only and includes all N losses (DIN, DON, N2, N2O) this doesn’t seem like a fair comparison per comments above.

Response 15: As discussed earlier in Response 8, the NLI is a budget diagnostic, and closing the budget is not only fair, but critical to understanding overall N fluxes through the system. As described in Response 7, however, we do agree with your request for more information on the partitioning of N outputs between benign and pollutant forms and we have made significant modifications to address this.

P1, L42: “rapidly grown”?

Response 16: We modified “rapidly grown” to “sharply increased” (Page 1, Line 40).

P1,L47: Again, there is very little empirical evidence for BNF increasing with CO2. It makes sense theoretically as BNF increases to meet CO2 induced demand but just not supported by evidence. Also this citation (12) doesn't work as they don't really even address BNF.

Response 17: We removed the citation the reviewer pointed out, and edited text to imply that increased BNF since preindustrial times are driven primarily by agriculture (Page 1, Lines 48-49). This is also discussed in Response 10.

P2, L6: True, that field description of many N cycling processes are less frequent in the tropics especially BNF and N2 losses. However, there are hundreds of empirical studies on many dimensions of tropical N cycling.

Response 18: We modified the text the reviewer pointed out. Page 2, Lines 8-10 now emphasize that uncertainty in the fate of Nr in many terrestrial and freshwater ecosystems remains a major research and management challenge, particularly for tropical regions, which are expected to experience the most dramatic increases in anthropogenic Nr inputs and LULUC in the next few decades (McIntyre et al., 2009; Zhu et al., 2005).

P2,L8: What is this saying? Of course natural BNF has added significant N since preindustrial times, but there is little evidence that it has increased.

Response 19: We removed the sentence the reviewer pointed out. Page 2, Lines 10-12 now state that increasing anthropogenic Nr inputs to relatively N-rich tropical systems may result in more rapid N losses to the ocean and atmosphere than to N-limited temperate systems (Matson et al., 1999). Please also see Response 10 and Response 17.

P2, L27: "emission associated with harvest"...What? Also the English doesn't work. "Harvesting"? are these crops?

Response 20: The detailed description of the net harvest is provided in Responses 11.

P2,L33: This makes little sense. If the N2 losses are derived from haber-Bosch then yes, I see your point. However, most N2 losses from tropics for example likely do not but are natural.

Response 21: We removed the text the reviewer pointed out. As discussed earlier in Response 7, we have now significantly expanded our description and analysis of the sensitivity of our results to the partitioning land N outputs into environmentally benign vs. pollutant forms, and estimated both land N outputs and pollution to the ocean and atmosphere.

P3, L15: This index needs to be reevaluated. Given historically low dep and fert inputs into many tropical regions, of course the index is going to be sensitive given that all forms are included in losses while only Nr is included in inputs.

Response 22: As discussed earlier in Response 8 and Response 15, the NLI is a simple but informative budget diagnostic that compares total N outputs from the system against total N inputs to the system, as a ratio. Total land Nr inputs are the sum of simulated BNF, atmospheric deposition (Green et al., 2004), and synthetic fertilizers (Bouwman et al., 2013). Total land N outputs are the sum of river exports to the ocean, emissions to the atmosphere, and net harvest. All these N input and output terms are now more clearly described in Page 2, Lines 31-46.

P3, L23: What does this mean?

Response 23: For clarification, we modified the sentence, which is now in Page 4, Lines 41-43.

P3, L35: “63% of global harvest emissions”. Is this deforestation? What is this?

Response 24: The detailed description of the net harvest is provided in Response 11. For clarification, we also modified the sentence, which is now on Page 4, Lines 11-13.

Supplemental Page 1, L17: Vitousek actually reports much lower isotope based estimates for BNF (58 Tg N yr-1)

Response 25: We now compare simulated preindustrial vs. contemporary BNF in all lands, and contemporary BNF in agricultural vs. non-agricultural lands with the corresponding reported values including the estimate by Vitousek et al. (2013). These results are presented and discussed in detail in Supplementary Note 1 (Supplementary Information, Page 2, Lines 14-32).

S1,L21: Huh? No influence of different BNF levels do not have much influence? I get that is insensitive to low versus high but it is unclear how the literature range was incorporated in the simulations. Are these used to constrain preindustrial spin up or to match contemporary observations. The references used represent a mixture of preindustrial BNF estimates and contemporary. It is also worth looking into Cleveland et al (PNAS 2013) for satellite and modeling based estimates of contemporary biome BNF rates for unmanaged lands.

Response 26: We now explicitly separate preindustrial, contemporary, agricultural and non-agricultural BNF, which are discussed in Response 25.

S1, L26: Confusing

Response 27: We modified the text, which are now in Supplementary Information, Page 2, Lines 48-50, which we have pasted below for reference:

Applications of the both fertilizers show higher land N sequestration in the extratropics than in the tropics (Fig. 3a) and create similar global, extratropical, and tropical land N pollution and fluxes (Fig. 3a, Fig. 4, Table 1, Supplementary Fig. 3 and 8).

S1, L34: Again “harvest” needs more explanation

Response 28: We have clarified this in Response 11.

S1,L39: How N₂ was modeled needs much more clarification. For tropical forest denitrification (total N gas emissions) estimates see Bai et al. (Biogeosciences 2012) and Brookshire et al. (GRL 2017)

Response 29: LM3-TAN simulates soil and freshwater denitrification (N₂+N₂O) emissions, which is described in Methods (Page 8, Lines 16-18 and Lines 26-37) and Supplementary Information (Supplementary Note 4, Supplementary Table 3). We have now greatly enhanced our discussion of the treatment and partitioning these emissions into N₂ and N₂O as described in Response 7.

Reviewer #2 (Remarks to the Author):

This is a revised version of manuscript I reviewed before, which was submitted to Nature. Lee et al applied the GFDL Land model -Terrestrial and Aquatic Nitrogen (LM3-TAN) to analyze the effect of the past two and half centuries of anthropogenic Nr inputs, land-use and land-cover changes (LULCC), increasing atmospheric CO₂, and climate on global and basin-scale N fluxes to the oceans and atmosphere. Major conclusions include:1) Globally, land transitioned from a net N source to N sink in the late 1940s, largely due to enhanced vegetation growth under elevated atmospheric CO₂; 2) Tropical land has produced more than 50% of contemporary land N outputs despite covering only 34% of global land area and receiving far lower synthetic fertilizer applications than the extratropics. Expanding LULCC in tropical regions could shift the global land balance back to a net N source. This revised manuscript has addressed many of my concerns. Below are some comments needed to be addressed before publication.

We are glad that our new manuscript has addressed many of your previous concerns, and would like to further thank the reviewer for providing two rounds of constructive comments. We hope that our responses to this round of comments have fully addressed the reviewer's remaining concerns. In particular, we have taken numerous steps to fully incorporate our uncertainty analyses into the main text as requested. These steps are detailed in our responses to the reviewer's individual comments below.

1) As the authors stated, the factor driving land from N source into N sink is elevated CO₂ concentration. The sensitivity analysis of removing CO₂ fertilization effect delays such a global land transition from a N source to N sink (Fig 2b). Thus, the accurate simulation of nutrients limits to the CO₂ effect on vegetation growth and the consequent biomass accumulation is critical. Large uncertainty exists in CO₂ fertilization effect on vegetation growth, particularly there is lack of observational evidence to quantify the magnitude of CO₂ fertilization effect on vegetation growth and BNF. It's necessary to have a discussion on this uncertainty in the manuscript.

Response 1: We have now discussed the uncertainty in CO₂ fertilization effects and its effect on our primary findings throughout the manuscript (Page 3, Lines 14-16 and 44-46; Page 4, Lines 13-14 and 32-39; Page 6, 15-19). The “no CO₂-fertilization” case is included in key figures (Fig. 2, 5 and 7) and we explicitly demonstrate the robustness of our conclusion to this case (Supplementary Fig. 4).

2) It has been long recognized that tropical forests' growth are largely limited by phosphorus (P) availability. I noted that TEM –C-N model used in LM3_TAN does not account for P

limits, which can potentially overestimate CO₂ impacts on C and N cycling. A discussion on this uncertainty is also needed.

Response 2: We now discuss the implications of P limitation in the tropics on Page 7, Lines 4-13. While this is a notable limitation of LM3-TAN, the net effect of P limitation in the tropics would not modify our conclusion – the prominence of the tropics as a source of global N pollution. We have pasted the newly added discussion of the implications of P limitation in the tropics on Page 7, Lines 4-13 below for reference:

An important factor that is not resolved in our model is the potential role of phosphorous (P) in limiting tropical production. We would expect that BNF in response to CO₂ fertilization is particularly restricted in P-limited tropical intact forests⁷⁰. We would not expect, however, that the BNF response is restricted in mostly N-limited tropical regrowing and temperate forests⁷¹⁻⁷² or agricultural lands. Like the “no CO₂ fertilization” case, overall N outputs would be ultimately most strongly shaped by accelerating LULUC. That is, despite the restricted BNF in tropical intact forests, overall tropical N outputs would be only modestly reduced, as the output fluxes would be supplied through release of legacy land N storage. Indeed, the P-limited tropics combined with enhanced land N sequestration in the N-limited and CO₂ fertilized extratropics would likely accentuate the prominence of the tropics as a source of global N pollution.

3) *Some key indicators such as NLI need statistical test for significance.*

Response 3: As the reviewer suggests, we have now implemented the uncertainty analysis more fully into the text. We have clearly described the baseline and uncertainty simulations in Page 3, Lines 6-16, and reported our results with uncertainty ranges throughout the paper (e.g., Table 1 and Supplementary Table 1). For NLI, we now report the standard deviation of the NLI estimates between a range of BNF settings, different fertilizer inputs, different LULUC scenarios (Fig. 3c) (Page 4, Lines 51 to Page 5, Lines 1-3).

4) *If you have implemented a set of simulations with different input data such as BNF settings, different fertilizers, different LULCC scenarios, etc, I suggest you report your results as mean+/-deviation.*

Response 4: As mentioned in *Response 3*, we have now included uncertainty bounds for all key quantities (Table 1, Supplementary Table 1, Fig. 2 vs. Supplementary Fig. 3, Fig. 3, Fig. 4 vs. Supplementary Fig. 4, Fig. 5 vs. Supplementary Fig. 5-8).

5) *A discussion on uncertainty analysis should be a necessary component of this work, which should be added to the main body of this manuscript.*

Response 5: We agree, and hope that the changes that were made in response to your comments above has addressed this concern. Specifically, we made the following changes throughout the manuscript:

- We have expanded our uncertainty analyses (Page 2 48-51 to Page 3, Lines 1-4) and added a clear description paragraph of the baseline and uncertainty simulations (Page 3, Lines 6-16).
- We have included uncertainty bounds for all key quantities (Table 1, Supplementary Table 1, Fig. 2 vs. Supplementary Fig. 3, Fig. 3, Fig. 4 vs. Supplementary Fig. 4, Fig. 5 vs. Supplementary Fig. 5-8).
- We have expanded discussions of key uncertainties, including uncertainties in CO₂ fertilization (See *Response 1*) and the potential role of P limitation on tropical production (See *Response 2*).
- In the new Fig. 4, we demonstrate that the robustness of our conclusion concerning the prominence of the tropics as a source of global N pollution to 1) the uncertainty in the different BNF, fertilizer inputs, and LULUC scenarios, 2) the different partitioning of land N outputs into environmentally benign vs. pollutant forms, and 3) the exclusion of river DON exports from land N pollution. Supplementary Fig 4 extends this further by demonstrating the robustness of the results to the complete exclusion of CO₂ fertilization.

REVIEWERS' COMMENTS:

Reviewer #1 (Remarks to the Author):

I reviewed a previous version of this manuscript. The authors have addressed many of my concerns. There remain a few minor editorial changes that should be made. Further, it remains unclear how biological N fixation has responded to CO₂ fertilization and land use in non-agricultural lands in the Amazon.

P1,L45: awkward and unclear sentence.

P2, L31-: Much of the first and third paragraphs read like they should be in the Methods not results.

P3, L19: How exactly was "consistency" determined. Maybe I missed this in the methods but "consistent" is vague and imprecise. This should be evaluated statistically.

P4, L19: This sentence and the use of "thus" does not follow logically from previous sentence.

P6, L39: While I see that BNF in non-agricultural lands have not changed much across the global tropics (Fig. 5), I remain unconvinced that BNF in non-agricultural lands of the Amazon have increased by ~2 Tg N /yr over the Anthropocene (Fig. 7) with CO₂ fertilization. There is little empirical evidence for this and unclear how this emerges in model simulations.

Reviewer #2 (Remarks to the Author):

I found the authors have addressed all my comments carefully, and particularly made a lot effort in uncertainty analysis. Estimates on soil N storage and fluxes are far from certain, showing large divergence among land biosphere models as indicated in a recent model ensemble study (Tian et al. GCB 2019). In the future work, therefore, multi-model intercomparison would be needed to further identify uncertainty associated model structure and hence improve model simulation for N fluxes and storage. Although Much uncertainty still exists in the current analysis, I think the work presented in this manuscript has made an important contribution to the field. Thus I would like to welcome this paper appearing in the literature.

Note: LULUC (Land Use and Land Use Change) is uncommon, suggesting you use LULCC (land use and land cover change) or LCLUC (Land cover and land use change).

Reference:

Tian, H., J. Yang, R. Xu, C. Lu, J.G. Canadell, E.A. Davidson, R.B. Jackson, A. Arneeth, J. Chang, P. Ciais, S. Gerber, A. Ito, F. Joos, S. Lienert, P. Messina, S. Olin, S. Pan, C. Peng, E. Saikawa, R.L. Thompson, N. Vuichard, W. Winiwarter, S. Zaehle, B. Zhang (2019) Global soil nitrous oxide emissions since the preindustrial era estimated by an ensemble of terrestrial biosphere models: Magnitude, attribution, and uncertainty, *Glob Change Biol.* 25: 640–659.
<https://doi.org/10.1111/gcb.14514>

Reviewer #1 (Remarks to the Author):

I reviewed a previous version of this manuscript. The authors have addressed many of my concerns. There remain a few minor editorial changes that should be made. Further, it remains unclear how biological N fixation has responded to CO₂ fertilization and land use in non-agricultural lands in the Amazon.

We are glad that our previous revised manuscript has addressed many of the reviewer's previous concerns and would like to further thank the reviewer for providing additional helpful comments. We hope that our responses to this round of comments have fully addressed the reviewer's remaining concerns. The modifications are detailed in our responses to the reviewer's individual comments below.

P1, L45: awkward and unclear sentence.

We modified the sentence to make it clear, which is now on P1L36-37 as follows:
Much attention has been placed on the effects of sharply increased anthropogenic Nr inputs on severe oceanic or atmospheric pollution^{5,8-10}.

P2, L31-: Much of the first and third paragraphs read like they should be in the Methods not results.

We agree that much of the text in the paragraphs belongs to Methods. We therefore moved it into Methods (which are now on P7L38-39, P8L3-4, and P9L18-38), except several sentences (P2L24-31 and P3L31-33) required for the reader to follow the paper's narrative. Accordingly, we direct to Methods when the moved text is first mentioned in the main text (P2L31-32, P3L9-10, and P3L33-34) to help readers' understanding.

P3, L19: How exactly was "consistency" determined. Maybe I missed this in the methods but "consistent" is vague and imprecise. This should be evaluated statistically.

We consider consistency as being within the uncertainty bounds of previously published estimates (e.g. Supplementary Table 1 and Supplementary Note 1). Please see P9L18-38 for a detailed description of our uncertainty analyses. We removed "consistency" and modified the sentence to clarify how we evaluated our results (P2L34-36) as follows:
Simulated global land N storage and fluxes in LM3-TAN are found to be within published uncertainty bounds in 16 different studies, when comparable categorization, definitions, and assumptions are applied (Figure 1, Supplementary Table 1, Supplementary Note 1).

P4, L19: This sentence and the use of “thus” does not follow logically from previous sentence.

We added a sentence between the two sentences to use “thus” logically. Now the three sentences are on P3L40-43 as follows:

In the tropics, the largest contributors to increasing land N outputs are net harvest and denitrification (Figure 5c). As mentioned previously, the majority of the net harvest is assumed to ultimately go to the atmosphere^{1,6,8}. Most of the increasing tropical N outputs thus go to the atmosphere, and river exports to the ocean remain relatively stable.

P6, L39: While I see that BNF in non-agricultural lands have not changed much across the global tropics (Fig. 5), I remain unconvinced that BNF in non-agricultural lands of the Amazon have increased by ~2 Tg N /yr over the Anthropocene (Fig. 7) with CO₂ fertilization. There is little empirical evidence for this and unclear how this emerges in model simulations.

We agree that there is little empirical evidence for BNF in intact forests increasing with CO₂ fertilization. We thus added a paragraph in P6L24-31 to note that there remains large uncertainty in our simulations as follows:

Our results, those of Brienen and colleagues⁵³, and other bottom-up estimates³⁷ are indicative of enhanced growth and C demand in intact forests, hypothesized to result from CO₂ fertilization³⁵. In LM3-TAN, BNF responds dynamically to enhanced C demand²⁸, and thus increases particularly in Amazonian non-agricultural lands including large intact forests (Figure 7b). Although this result is plausible, it should be noted that large uncertainty remains in these simulations with little empirical evidence for BNF in intact forests increasing with CO₂ fertilization. While our conclusions concerning the contribution of tropical systems to global N pollution is insensitive to this response, further observational work and investigation with multi-model ensembles⁵⁴ could help resolve this uncertainty.

Reviewer #2 (Remarks to the Author):

Reviewer #2 (Remarks to the Author):

I found the authors have addressed all my comments carefully, and particularly made a lot effort in uncertainty analysis. Estimates on soil N storage and fluxes are far from certain, showing large divergence among land biosphere models as indicated in a recent model ensemble study (Tian et al. GCB 2019). In the future work, therefore, multi-model intercomparison would be needed to further identify uncertainty associated model structure and hence improve model simulation for N fluxes and storage. Although Much uncertainty still exists in the current analysis, I think the work presented in this manuscript has made an important contribution to the field. Thus I would like to welcome this paper appearing in the literature.

Note: LULUC (Land Use and Land Use Change) is uncommon, suggesting you use LULCC (land use and land cover change) or LCLUC (Land cover and land use change).

Reference:

*Tian, H., J. Yang, R. Xu, C. Lu, J.G. Canadell, E.A. Davidson, R.B. Jackson, A. Arneeth, J. Chang, P. Ciais, S. Gerber, A. Ito, F. Joos, S. Lienert, P. Messina, S. Olin, S. Pan, C. Peng, E. Saikawa, R.L. Thompson, N. Vuichard, W. Winiwarter, S. Zaehle, B. Zhang (2019) Global soil nitrous oxide emissions since the preindustrial era estimated by an ensemble of terrestrial biosphere models: Magnitude, attribution, and uncertainty, *Glob Change Biol.* 25:640–659. <https://doi.org/10.1111/gcb.14514>.*

We are glad that our previous revised manuscript has fully addressed the reviewer's concerns, and would like to thank the reviewer again for providing us a number of constructive comments on this and very early versions of this manuscript which we believe has helped to improve our manuscript substantially.

We agree that there remains substantial uncertainty in land N storage and fluxes and thus, if any, would be eager to participate in a multi-model intercomparison study in the near future. In P6L24-31, we have added discussion recognizing the value of multi-model ensemble analyses for resolving remaining land N budget uncertainties with the suggested reference.

Finally, as the reviewer suggested, we modified LULUC to LULCC throughout the manuscript.